# The Effect of MC-Type Carbides on the Microstructure and Wear Behavior of S390 High-Speed Steel Produced via Spark Plasma Sintering

Qipeng Hu [1,2,3], Miaohui Wang [1,2,3,*], Yunbo Chen [1,2], Hailong Liu [2,3,*] and Zhen Si [1]

1 China Academy of Machinery Science and Technology Group Co., Ltd., Beijing 100044, China
2 Beijing National Innovation Institute of Lightweight Co., Ltd., Beijing 100083, China
3 China Machinery Institute of Advanced Materials Co., Ltd., Zhengzhou 450001, China
* Correspondence: wangmh0103@163.com (M.W.); liuhailo17@tsinghua.org.cn (H.L.)

**Abstract:** The microstructure and wear behavior of S390 high-speed steel (HSS) reinforced with different volume fractions of MC-type carbides produced via spark plasma sintering were investigated using scanning electron microscopy (SEM) and transmission electron microscopy (TEM) in this study. SEM and TEM results show that V-W-rich carbides are formed around the added MC-type carbides, and these carbides have a similar composition to the M(C, N) carbides precipitated at high temperatures according to thermodynamic calculations. Both macrohardness and three-point bending results show that the carbide type is the dominant factor increasing the hardness, and the volume fraction of the carbide is the dominant factor leading to a decrease in the three-point bending strength. The wear mechanism of HSS metal matrix composites (MMCs) is confirmed as abrasive wear and oxidative wear via wear tracks and oxidation films. Compared with the sample without reinforcement (85 HRA, wear coefficient of $1.50 \times 10^{-15}$ m$^2$/N), the best MT-3 sample exhibits a hardness increase of 1.8 HRA and a three-fold increase in wear resistance.

**Keywords:** metal matrix composites; high-speed steel; MC-type carbides; microstructure; wear





## 1. Introduction

High-speed steels (HSSs) are considered to be an ideal candidate material for high-performance rollers, complex gear cutters, fine blanking dies, and high-temperature bearings due to their high hardness and excellent wear resistance under various wear conditions [1–4]. Generally, HSSs are mainly produced via conventional casting and forging and emerging powder metallurgy (PM). Compared to the former, PM can produce high-alloy HSSs characterized by finer carbides uniformly distributed in a martensitic matrix, which results in higher toughness and wear resistance [5–8]. In addition, PM, such as hot isotopic pressing (HIP), vacuum sintering (VS), powder injection molding (PIM), and spark plasma sintering (SPS), is also commonly used for the preparation of numerous metal matrix composites (MMCs) [9–12].

MMCs have shown some special advantages in hardness, toughness, and wear resistance due to the integration of matrix and ceramic particles. Notably, the microstructure and mechanical properties of MMCs are deeply dependent on metal matrix and additional ceramic particles. Moreover, these ceramic particles generally have two categories: inertia particles and reactive particles. Numerous MMC studies focus on the inertia particles with good thermodynamic stability that are hardly decomposed during in-service life, such as VC, WC, TiC, NbC, TaC, and Al$_2$O$_3$. Herranz et al. [13] reported that the hardness of M2 HSSs reinforced with VC produced using concentrated solar energy sintering ascended with sintering temperature and the addition of VC. However, the number density of additive carbides is an influential factor responsible for wear properties, rather than

macrohardness, in the M3/2 HSS reinforced with NbC and TaC via cold isostatic pressing [14]. Zhang et al. [15] made a comparison between in situ and ex situ WC-reinforced iron matrix composites using spark plasma sintering, which demonstrated that the wear resistance of the former surpassed that of the latter. Othsuka et al. [16] found that the addition of B and $Al_2O_3$ greatly improved the hardness of PIMed SKH51 HSSs, reaching a hardness of 1000 Hv. Table 1 illustrates the reported experimental results of various MMCs strengthened using ceramic particles, which shows the clear effect of the type and volume fraction of ceramic particles on hardness.

Some researchers have investigated the in situ particles produced by the reaction between matrices with additive ceramic particles, such as SiC and $TiB_2$. Fedrizzi et al. [17] observed the formation of TiC and $Fe_2B$ shells around $TiB_2$ particles in the H13 tool steel with $TiB_2$, and these in situ particles were qualitatively analyzed as a reaction product of $TiB_2$ and $\alpha$-Fe. Wu et al. [18] also reported that the microhardness of 316L stainless steels produced using laser melting deposition increased to 974 Hv due to the addition of SiC, and found that SiC reacted with the iron matrix to generate the $Fe_3Si$ and FeSi.

**Table 1.** The reported experimental results of various MMCs.

| Matrix | Consolidation | Particle | Addition (vol.%) | Hardness (HRA) | Ref. |
|--------|---------------|----------|------------------|----------------|------|
| M2 | Tubular furnace sintering (1140–1295 °C) | VC | 0/3/6/10 | 79.3 */84.2 */84.6 */84.8 * | [13] |
| | Solar furnace sintering (940–1150 °C) | VC | 0/3/6/10 | 85.6 */85.75 */85.9 */86.35 * | |
| M3/2 | Vacuum sintering | TiC | 0/5 | 77.4 */78.5 * | [19] |
| | | MnS | 0/5 | 77.4 */74.7 * | |
| | | TiC + MnS | 0/5+5 | 77.4 */76.3 * | |
| | Sintering (1250 °C) | $CaF_2$ | 5 | 73.2 * | [20] |
| | | TiC | 5 | 68.4 * | |
| | | MnS | 5 | 63.8 * | |
| | | $CaF_2$ + TiC | 5+5 | 71.8 * | |
| | Vacuum sintering (1230 °C) | NbC | 0/5/7.5/10 | 79.6 */79 */79.6 */80.7 * | [14] |
| | | TaC | 0/5/7.5/10 | 79.6 */74.7 */79 */79 * | |
| | | NbC/TaC | 0/5/7.5/10 | 79.6 */79 */78.5 */77.4 * | |
| | Vacuum sintering (1150 °C) | WC | 0/10/30 | 65.3 */66.5 */68.2 * | [21] |
| 316L | Laser fusion | SiC | 0/3/6/9 | 62.4 */69 */72.5 */76.1 * | [22] |
| | Laser melting deposition | SiC | 4/8/12/16 | 68.9 */72.4 */82.45 */86.05 * | [18] |

*: microhardness and macrohardness values are converted to HRA hardness.

The PMHSSs with a martensitic structure are widely applied in high-performance gear cutters and fine blanking dies due to their excellent wear resistance, hardness, good temper resistance, and comprehensive mechanical properties. Although MC-type carbides (TiC, WC, VC, NbC, and SiC) are widely used as a ceramic reinforcement of MMCs, previous studies mainly focus on the effect of the volume fraction of a single MC-type carbide on the mechanical properties, wear resistance, and microstructures of HSS MMCs. However, the synergistic effect of multiple carbides and carbide type can also affect the microstructure, hardness, fracture toughness, and wear properties of HSS MMCs. Moreover, the bonding strength of the interfaces between ceramic particles and matrices is much dependent on the chemical stability of extrinsic carbides. In this study, the effect of the type and volume fraction of MC-type carbides and their synergistic effect on the microstructure, mechanical properties, and wear resistance of HSS MMCs was investigated using scanning electron microscopy (SEM) and transmission electron microscopy (TEM). Meanwhile, the microstructure evolution of the precipitates around additive MC-type carbides and their influence on the bending strength and microstructure of fracture surface were analyzed.

## 2. Materials and Experiments

### 2.1. Materials

As shown in Table 1, MC-type carbides, such as SiC, TiC, and VC, are widely used to produce MMCs with excellent wear resistance and toughness. Therefore, three MC-type carbides (SiC, TiC, and VC) were selected as extrinsic reinforcement particles in this study. Figure 1 shows the SEM micrographs of a S390 HSS nitrogen-atomized powder and three MC-type carbides used in this study. The chemical composition of the S390 powders and size distribution of the raw S390 and MC-type carbide powders are listed in Tables 2 and 3, respectively. As shown in Table 3, the size distribution of the S390 powders (5–54 μm) is wider than that of the carbide particles (1–20 μm). A certain range of the powder was chosen to avoid powders with irregular morphology, such as flaky and hollow powder.

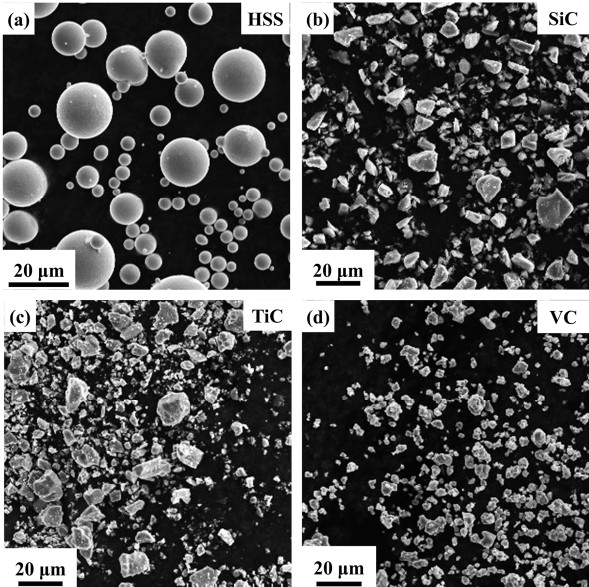

**Figure 1.** The SEM secondary electron (SE) micrographs of raw powders. (**a**) HSS, (**b**) SiC, (**c**) TiC, and (**d**) VC.

**Table 2.** The chemical composition of the S390 HSS powders.

| Element | C | Mo | W | Mn | Cr | V | Co | Fe |
|---------|------|------|-------|------|------|------|------|------|
| wt.% | 1.63 | 2.28 | 10.09 | 0.26 | 4.91 | 5.12 | 8.11 | Bal. |

**Table 3.** The size distribution of the raw S390 and MC-type carbide powders.

| Powder (μm) | D3 | D10 | D25 | D50 | D75 | D90 | D98 |
|-------------|------|------|-------|-------|-------|-------|-------|
| S390 | 5.30 | 8.21 | 12.80 | 20.51 | 30.37 | 40.27 | 53.91 |
| SiC | 0.90 | 1.53 | 3.04 | 5.40 | 8.32 | 11.28 | 15.36 |
| TiC | 0.70 | 1.10 | 2.50 | 6.22 | 10.71 | 14.98 | 20.31 |
| VC | 0.74 | 1.33 | 2.72 | 5.83 | 9.41 | 13.01 | 16.22 |

Table 4 shows the volume fraction of MC-type carbides in various designed HSS MMCs. Different volume fractions of carbides were designed to investigate the desired materials. As shown in Table 1, the volume fraction of additional MC-type carbides was as high as 10 vol.% in prior reported studies. Due to the high volume fraction of intrinsic carbides in the S390 matrix, the volume fraction of extrinsic carbides was limited to below 7.5 vol.% to hold good toughness. To produce the homogeneously reinforced HSS MMCs, the raw powders were proportioned and blended before the consolidation process. The

powder mixing was conducted in a planetary ball mill (Chishun QM-3SP2, Nanjing, China), which can rotate in two directions. The mixture of S390 powders and carbide particles was further mixed for 2 h at 400 rpm with a ball-to-powder weight ratio of 7:1, and 316L stainless steel balls with a 5 mm diameter were employed to attain a uniform mixture.

**Table 4.** The volume fraction of MC-type carbides in various designed HSS MMCs.

| Sample | Base | Carbides (vol.%) | | |
| --- | --- | --- | --- | --- |
| | | SiC | VC | TiC |
| M | 100 | | - | - |
| MS-1 | - | 2.50 | - | - |
| MS-2 | - | 5.00 | - | - |
| MS-3 | - | 7.50 | - | - |
| MV-1 | - | - | 2.50 | - |
| MV-2 | - | - | 5.00 | - |
| MV-3 | - | - | 7.50 | - |
| MT-1 | - | - | - | 2.50 |
| MT-2 | - | - | - | 5.00 |
| MT-3 | - | - | - | 7.50 |
| MSV-1 | - | 1.25 | 1.25 | - |
| MSV-2 | - | 2.25 | 2.25 | - |
| MSV-3 | - | 3.25 | 3.25 | - |
| MST-1 | - | 1.25 | - | 1.25 |
| MST-2 | - | 2.25 | - | 2.25 |
| MST-3 | - | 3.25 | - | 3.25 |

### 2.2. SPS Method and Heat Treatments

The consolidation process was performed using the Dr. Sinter SPS machine (Sumitomo SPS-1050, Japan). The well-mixed powders were filled in a graphite die with an inner diameter of 30 mm. A graphite foil was inserted between the mixed powders and the die to facilitate demolding and extend the die's lifetime. The optimal parameters (a sintering temperature of 1050 °C, a heating rate of ~90 °C·min$^{-1}$, a uniaxial pressure of 50 MPa, and a holding time of 5 min) were used to prepare the S390 HSSs with MC-type carbides in the non-vacuum atmosphere. The sintered specimens were produced with a diameter of 30 mm and a thickness of 14 mm.

Figure 2 shows the heat treatment curves of SPSed MMC samples. The SPSed MMC specimens were subjected to two-step heating (550 °C and 850 °C, ~5 min) to reduce the cracking tendency before then being oil-quenched to room temperature (25 °C) and finally austenitized at 1150 °C. The decarburization surfaces were removed using a wire cutting. The triple tempering treatment was conducted at 550 °C for 1 h, followed by air cooling to room temperature (25 °C).

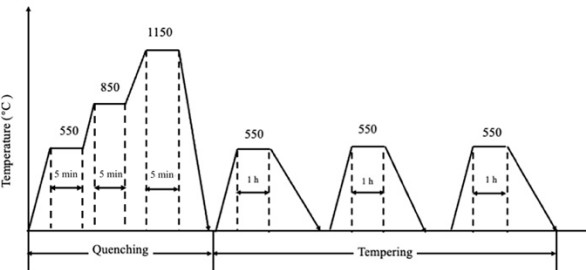

**Figure 2.** The schematic diagram of heat treatment of the S390 MMCs.

### 2.3. Microstructure Characterization and Mechanical Tests

The SEM samples were ground and polished in the conventional method before the SEM observation, and TEM samples were thinned using a GATAN model 695 ion mill

before TEM observation. The morphology and microstructure of designed MMCs were observed using a Zeiss Gemini SEM 500 scanning electron microscope (SEM) operating at 5 kV and a FEI Tecnai G2 transmission electron microscope (TEM) operating at 200 kV. The phase composition in the designed MMCs was analyzed using a TTRAX3 X-ray diffraction (XRD) technique with Cu Kα radiation. The wear tracks on the surface of worn samples were observed using SEM to clarify the distribution of the solute elements and the wear mechanisms. Moreover, JMatPro software was used to calculate the phase diagrams to investigate the chemical composition variation in precipitated carbides from 20 to 1600 °C with a step size of 10 °C.

Mechanical characterization was tested on polished specimens using Rockwell hardness measurements with a load of 1470 N for 14 s. For each specimen, 7 random indentations were made and averaged to obtain a correct hardness. Specimens with dimensions of $3 \times 6 \times 30$ mm$^3$ were cut from the designed samples to apply the three-point bending strength tests, and at least 3 samples were performed for every designed sample.

The density of the prepared HSS MMCs was measured using Archimedes' method. The relative density values were determined based on the rule of mixtures, and the volume loss of the pin was calculated from Equation (1). The wear coefficients κ of the designed samples was calculated based on Equation (2). Equations (1) and (2) are shown below:

$$V_{loss}\left(\mathrm{m}^3\right) = \frac{m_{loss}(\mathrm{g})}{\rho_{measure}(\mathrm{g}\cdot\mathrm{m}^{-3})} \tag{1}$$

$$\kappa_{wear}\left(\mathrm{m}^2\cdot\mathrm{N}^{-1}\right) = \frac{V_{loss}(\mathrm{m}^3)}{(F_n(\mathrm{N}) \times L_{sliding}(\mathrm{m}))} \tag{2}$$

The wear and friction tests were performed on a pin-on-disk MMU-10G test machine. The pin samples with a dimension of $\Phi 4 \times 15$ mm$^3$ were cut from the designed MMCs using a wire-cut spark machine. The HIPed S390 HSS were used as counterpart disks with a dimension of $\Phi 43 \times 3$ mm$^3$. The surfaces of pin and disk samples were polished with SiC abrasive papers from 240# to 1200#. The four-wear test was performed with a load of 1000 N and a rotation speed of ~400 rpm for each designed sample at room temperature. The experimental details of the wear test can be found in Table 5.

**Table 5.** The experimental parameters of the pin-on-disk wear test.

| Load/N | Sliding Speed/r min$^{-1}$ | Test Duration/min | Sliding Distance/m | Track Radius/mm |
|--------|----------------------------|-------------------|--------------------|-----------------|
| 1000   | 400                        | 20                | 553                | 11              |

## 3. Results and Discussions

### 3.1. Densification and Microstructures

Density values were obtained via Archimedes' method for the M sample and other designed MMC samples. The theoretical density for the S390 matrix was 8.2 g/cm$^3$, and the relative density was calculated following the rule-of-mixture equation.

Figure 3 illustrates the density and relative density of the designed S390 MMCs with an increasing volume fraction of carbides. A significant downward trend was observed in the density of all designed samples with the increasing volume fraction of carbides, as shown in Figure 3a. However, the relative density was above 97% for all samples, which implies that the SPS parameters used in this study could produce HSS MMCs with nearly full density. For carbide-reinforced HSS MMCs, the decrease in relative density was mainly due to the cluster of carbide particles [23,24]. By using S390 powders with a small size of D50 = 20.51 μm, the size and density of pores in designed MMC samples were reduced, which suppresses the cluster of carbide particles. Compared to the SiC-reinforced samples, the relative density of the samples reinforced with VC and TiC exhibited a continuous downward trend with increasing carbides. Conversely, the addition of SiC could elevate

the relative density of the S390 MMCs. As shown in Figure 3b, the relative density of the samples reinforced with SiC exceeded 99%.

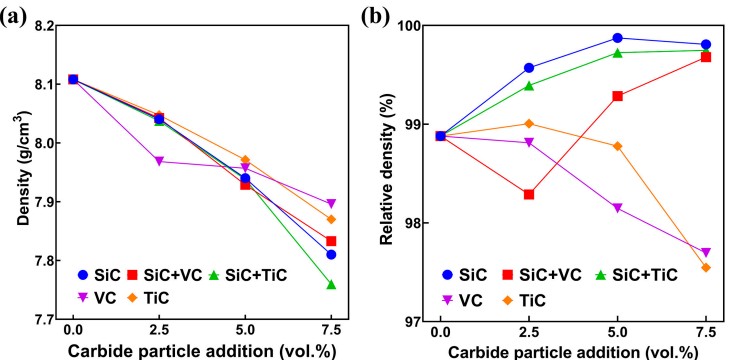

**Figure 3.** (**a**) The measured density and (**b**) calculated relative density of the S390 MMCs with an increasing volume fraction of carbides.

The SEM micrographs of the S390 MMCs with various carbides are shown in Figure 4. Fewer sintering pores and carbide clusters can be observed in MMC samples. Generally, the density and size of the pores formed during the sintering process had a major influence on the mechanical properties of the PM products. Compared with the control M sample (Figure 4a), some black or dark gray carbides with a diameter of ~5 µm were uniformly distributed around the powder boundary in carbide-reinforced MMC samples (Figure 4b–f). Figure 5 shows the XRD patterns of the S390 MMCs designed in this study. As shown in Figure 5, the M sample mainly exhibited five phases: $\alpha$-Fe, VC, $Fe_3W_3C$, $Cr_{23}C_6$, and $Cr_3C_2$. The additional MC carbides were also identified in the XRD patterns of other samples. However, the XRD diffraction peaks of TiC, SiC, and VC were too close to be clearly identified. The presence of the FeSi phase in SiC-reinforced MMC samples reveals that the SiC decomposed during the SPS process or heat treatment. The $V_2C$ phase from VC powder was also identified in MV-1 and MSV-1 samples. Previous studies also reported that SiC decomposed above 610 °C and formed $Fe_3C$ and $FeSi_x$ in SPS and the laser-melting deposition process [18,22,25]. Therefore, a rise in the relative density of the S390 MMCs with SiC can be attributed to the decomposition of SiC, which reduced the size of the sintering pores.

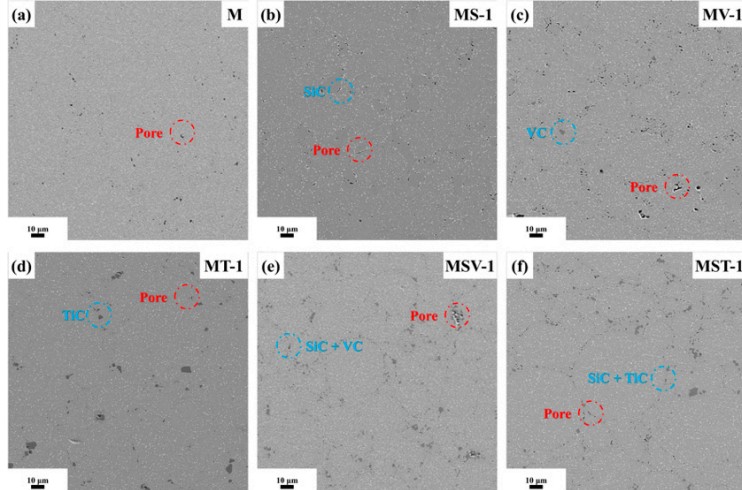

**Figure 4.** The SEM-SE micrographs of the S390 MMCs with different volume fractions of carbide. (**a**) M; (**b**) MS-1; (**c**) MV-1; (**d**) MT-1; (**e**) MSV-1; (**f**) MST-1.

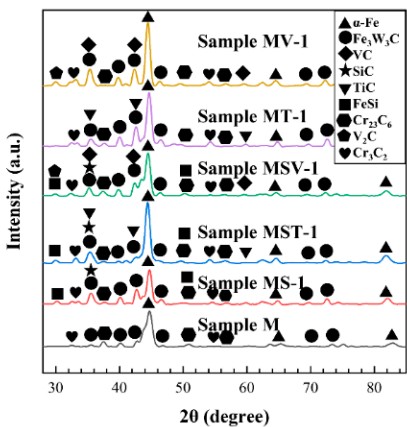

**Figure 5.** The XRD patterns of designed S390 MMCs designed in this study.

The SEM micrographs of the M sample and EDS results of the S390 MMCs are presented in Figure 6. As shown in Figure 6a, dark-gray and light-white particles were observed, which were V-rich MC carbides and W-rich $M_6C$ carbides, respectively. The thermodynamic equilibrium of S390 HSSs was predicted using JMatPro. Additionally, the calculated equilibrium phase diagram is exhibited in Figure 6b. According to Figure 6b, the four major phases ($\alpha$-Fe, MC, $M_6C$, and $M_{23}C_6$) were identified at room temperature. The thermodynamic calculation results corresponded with the experimentally observed carbide types shown in Figure 6a. The variations of elements were also investigated at various temperatures, and their results are shown in Figure 6c,d. As shown in Figure 6c, the MC carbide that existed at high temperatures had ~28 wt.% W elements, i.e., much higher than the ones that existed at room temperature. As a result, the MC carbide precipitated during the cooling process was mainly the V-rich MC carbide. The compositions of $M_6C$ carbides were also compared at different equilibrium temperatures. As shown in Figure 6d, the W content of $M_6C$ carbides at room temperature was approximately 17% higher than that at high temperatures. Therefore, the precipitation during cooling was mainly W-rich $M_6C$.

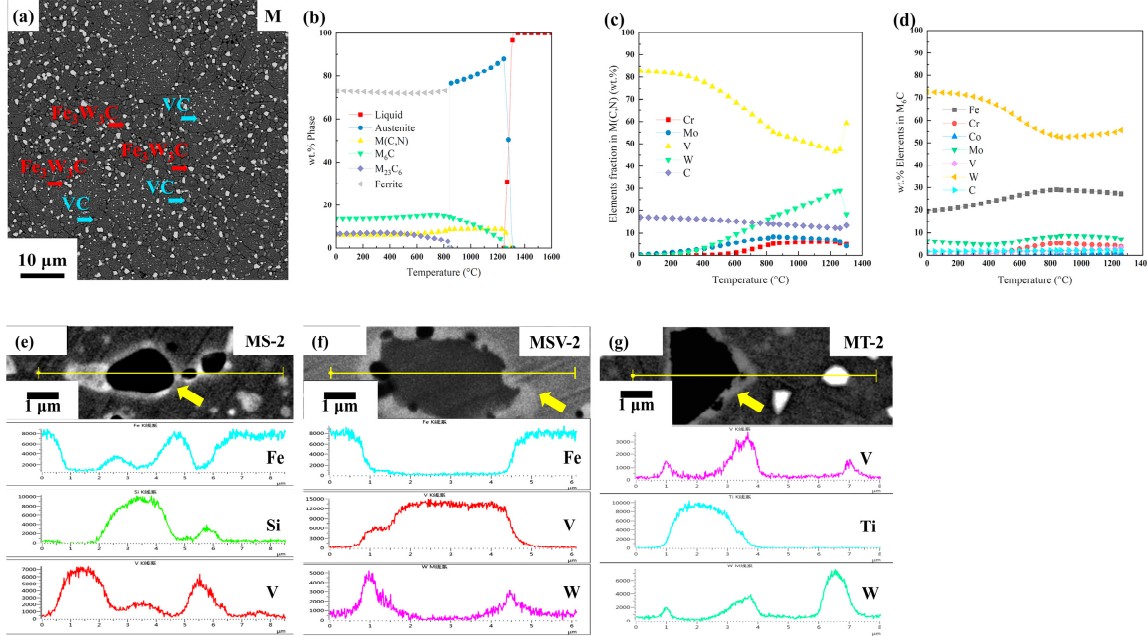

**Figure 6.** (**a**) The SEM backscattered electron (BSD) micrographs of the M sample, (**b**) calculated equilibrium phase diagram of S390 HSSs, (**c**) element variation in M (C, N), and (**d**) $M_6C$ predicted using JMatPro. The SEM-BSD micrographs and EDS results of S390 MMCs with different carbides: (**e**) MS-2; (**f**) MSV-2; (**g**) MT-2.

As shown in Figure 6e–g, the boundary between added MC-type carbides and the S390 matrix was surrounded by a carbide layer with similar chemical composition. In the MS-2 sample reinforced with SiC (Figure 6e), the SEM observation shows that there was a white layer and particle-type precipitates around the black particles. The EDS results also show that the white phase referred to V-rich MC-type carbides and the black phase referred to SiC particles. Compared to TiC and VC, the morphology of SiC particles transformed from an irregular shape to a pebble shape, which implied a reaction between SiC particles and the S390 matrix. The V-rich MC carbides that precipitated in the vicinity of SiC particles hindered further progress in the decomposition reaction. The decomposition and refinement stages of SiC were also observed in the laser fusion 316L MMCs, and the morphology of SiC exhibited a drop from micron sizes to near-nanometer sizes [22]. The white precipitates were also observed in the samples reinforced with VC (Figure 6b). Furthermore, the EDS results show that the white phase had a high content of W, which means that the phase around the VC particle belonged to W-rich $M_6C$ carbides. For the sample reinforced with TiC (Figure 6c), high contents of W and V were detected in the gray phase and the content of Ti was identified in black particles, which implies that the (V, W)-rich phase precipitated at the brim of TiC particles during the sintering and heat treatment process.

To clarify the characteristics and formation mechanisms of precipitates around the MC-type carbides, a 3 μm TiC particle was observed via TEM, which is shown in Figure 7. The selected area electron diffraction (SAED) pattern (Figure 7a) revealed that the gray precipitate belonged to the $VWC_2$ phase (ICSD #619038). The EDS analysis of the TiC particle (spot. 1), the iron matrix (spot. 2), and the $VWC_2$ gray precipitant (spot. 3) is illustrated in Figure 7b. Notably, the EDS analysis of spot. 3 confirms that these precipitates were associated with the (V, W, Mo)-rich carbide, and the composition of gray precipitates was consistent with the calculated composition of M(C, N) in high-temperature precipitation conditions. Therefore, the precipitation process can be promoted by a special sintering process during SPS, which creates spark plasma and a high temperature between the MC carbide particles and the S390 powder. From the EDS mapping results in Figure 7c–f, the discrete precipitation of (V, W, Mo)-rich carbides was distributed along the boundary of TiC, and the presence of the carbide was expected to create adhesion between a S390 matrix and a TiC particle. Similar observations were found in prior studies, which reported that TiC was surrounded by the V-rich MC carbide and that (W, Mo)-rich $M_6C$ was segregated at grain boundaries in sintered TiC/M2 HSS [26] and TiC/M3/2 HSS [19]. The addition and heat treatment of carbides had a significant impact on the formation and composition of precipitates between the MC carbide and the MMC matrix. Zhang et al. reported that controlling the cooling rate could alter the V or Ti content of (Ti, V, Mo)C particles in Ti–V–Mo micro-alloyed steel [27]. Wang et al. also reported that the shape of (Ti, Mo)C changed due to interface reactions during isothermal holding [28]. The simulation results show that (V, M)C (M = W, Mo, and Cr) were thermodynamically stable and the (Cr, Mo, W, Mn, V)-doped interface enhanced the ferrite/TiC interface [29–32].

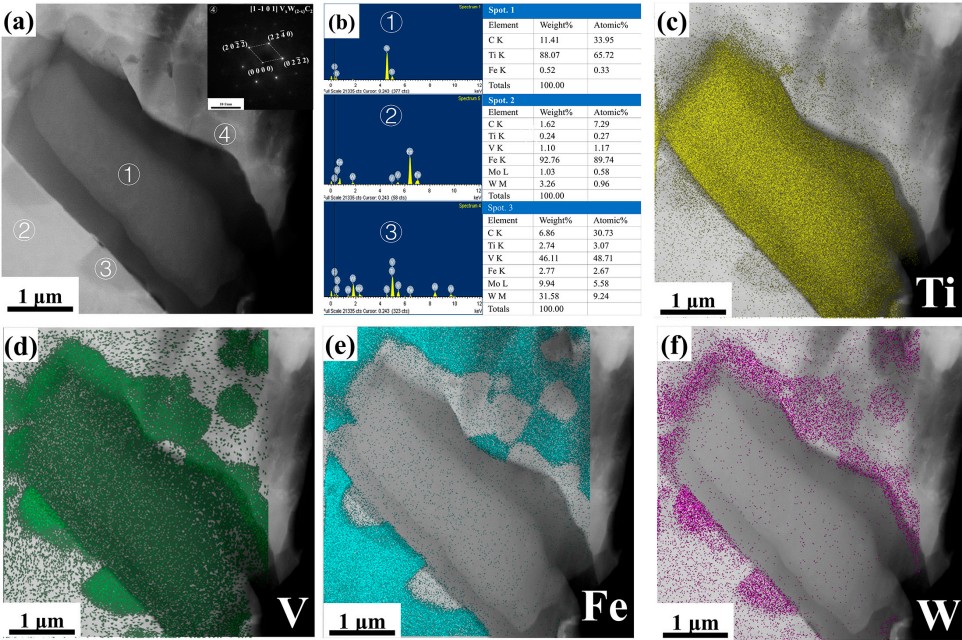

**Figure 7.** (**a**) The TEM images of TiC in the S390 MMCs and the SAED pattern of spot 4, (**b**) the corresponding EDS spectrum of 3 spots in (**a**), and (**c**–**f**) the EDS mapping.

## 3.2. The Mechanical Properties of the S390 MMCs

Figure 8 displays the relationship between the hardness and volume fraction of the MC-type carbides. Due to the high hardness of the matrix (85 HRA), the addition of various carbides exhibited limited hardness enhancement (Figure 8a), and the hardness of S390 MMCs was affected by the volume fraction and type of MC-type carbide. For the same carbides, the hardness enhancement was much affected by the volume fraction of additional carbides. As shown in Figure 8b, the addition of TiC and VC could help to improve the hardness of the S390 MMCs, and each additional 2.5 vol.% of TiC carbide particles could increase the hardness by ~0.6 HRA. Conversely, the improvement can be negligible in the S390 MMCs with SiC, and hardness values fluctuated in the range of 85 to 86 HRA. Notably, the addition of SiC + VC and SiC + TiC had minor improvements, and the addition of 7.5 vol.% of hybrid carbide particles only improved the hardness by ~1 HRA. The carbide type also had a significant effect on the hardness improvement due to the intrinsic characteristics of the MC carbide. As shown in Table 6, TiC was the highest, followed by VC, and finally SiC. Therefore002C TiC could prominently improve the hardness of HSS MMC. The samples with 7.5 vol.% TiC had the highest room-temperature hardness of 86.8 HRA.

**Table 6.** The intrinsic characteristics of carbides.

| Compound | Density (g/cm$^3$) | Vickers Hardness (GPa) | Ref. |
|---|---|---|---|
| Fe$_3$W$_3$C | 14.3 | 16.8 | [33] |
| SiC | 3.2 | 24.5–28.2 | |
| VC | 5.8 | 27.2 | [34] |
| TiC | 4.9 | 28.0–35.0 | |

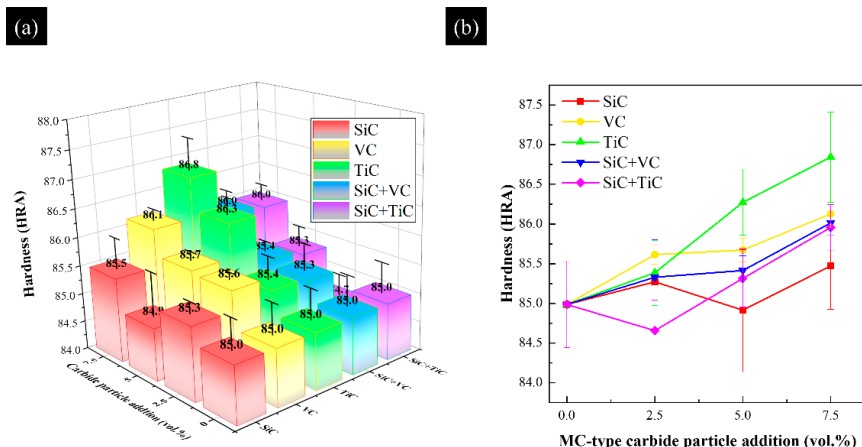

**Figure 8.** The effect of the volume fraction and type of additive MC-type carbides on the room-temperature hardness of designed MMCs. (**a**) Histograms of hardness vs. volume fraction and type of carbides; (**b**) Curve of hardness vs. MC-type carbides addition.

The bending strength of designed samples was investigated using a three-point bending test, as shown in Figure 9. For all samples, the control M sample demonstrated the highest bending strength of ~1197 MPa (Figure 9a). Except for samples reinforced with SiC, the addition of carbides deteriorated the bending strength with an increasing volume fraction of carbides. For example, Figure 9b shows that the bending strength of samples reinforced with TiC descended from ~1197 Mpa to ~899 Mpa with an increasing volume fraction of carbides. A similar phenomenon was also observed in the TiC- and TiN-reinforced M3/2 HSS [35]. The fracture surface analysis of three-point bending samples with 7.5 vol.% TiC and 7.5 vol.% VC is shown in Figure 10, revealing the effect of (V, W, Mo)-rich carbides on the fracture behavior of TiC carbides. As shown in Figure 10, some TiC and VC particles were torn rather than pulled out at the fracture surfaces. Moreover, the (V, W, Mo)-rich carbides outside the carbide particles exhibited the same behavior. Some prior studies found that cracks tended to initiate the incoherent interface between $M_6C$ and martensite rather than the semi-coherent interface between MC and martensite [36,37]. Therefore, the discrete precipitation behavior created an MC-(V, W, Mo)-rich carbide matrix structure, which might enhance the bonding of MC particles and matrices. However, the bonding effect played a small role in improving the three-point bending strength due to the irregular morphology of MC particle properties, which might cause stress concentration during servicing.

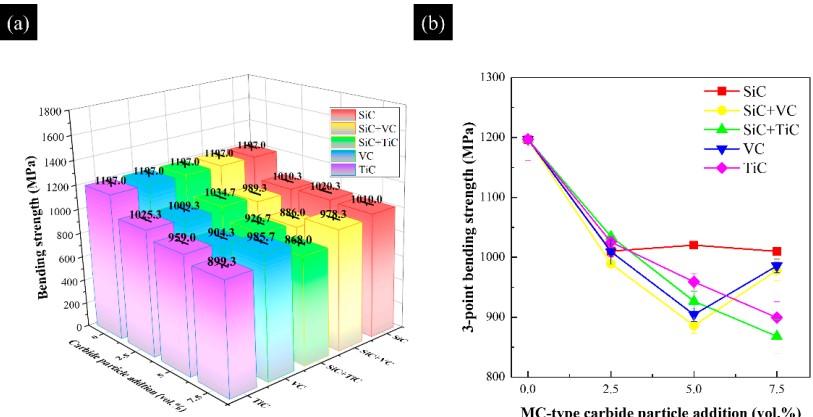

**Figure 9.** The effect of the volume fraction and type of additive MC-type carbides on the 3-point bending strength of designed MMCs. (**a**) Histograms of 3-point bending strength vs. volume fraction and type of carbides; (**b**) Curves of 3-point bending strength vs. MC-type carbides addition.

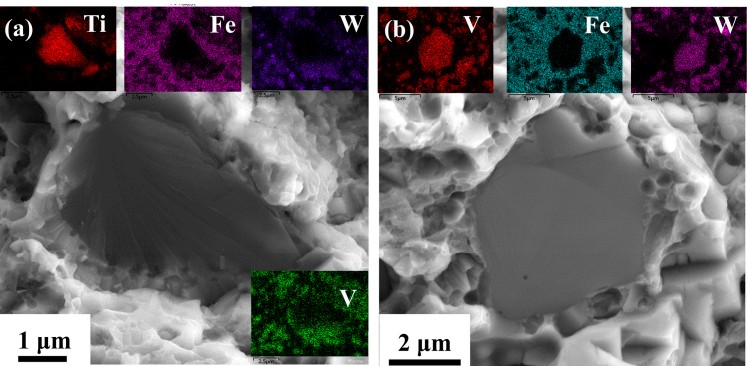

**Figure 10.** The SEM-BSD and EDS micrographs of the 3-point bending fracture surface of the S390 MMCs: (**a**) MT-3 and (**b**) MV-3.

### 3.3. Wear Test and Observations

Figure 11 illustrates the relationship between the volume fraction and type of added carbides and wear properties. As shown in Figure 11a, the M sample without the addition of carbides had the highest average wear coefficient of $1.5 \times 10^{-15}$ m$^2$/N. Conversely, the wear coefficients of designed samples reinforced with MC-type carbides were much lower than those of the M samples, which implies that the designed MMCs had better wear properties. As shown in Figure 11b, for the same carbides, the wear coefficients of MMCs exhibited a downward trend as the volume fraction increased. Compared to SiC-and VC-reinforced MMCs, the S390 MMCs reinforced with TiC had the best wear performance under the same volume fraction, and the wear coefficient decreased from $1.65 \times 10^{-15}$ m$^2$/N (MT-1 sample) to $4.64 \times 10^{-16}$ m$^2$/N (MT-3 sample, the lowest among the designed samples). With the volume fraction of SiC increasing, the wear coefficient of MMCs decreased from $1.23 \times 10^{-15}$ m$^2$/N (MS-1 sample) to $6.55 \times 10^{-16}$ m$^2$/N (MS-3 sample). The effect of the carbide type on wear resistance was also investigated in this study, which shows that the carbide type is a vital influencing factor in improving the wear properties: TiC > VC > SiC + TiC > SiC + VC > SiC. Moreover, the enhancement of wear resistance by carbides was consistent with the intrinsic hardness of carbides (Table 6). However, the effect of the carbide type on wear resistance was much less significant than that of the volume fraction, which is consistent with prior studies [14,38].

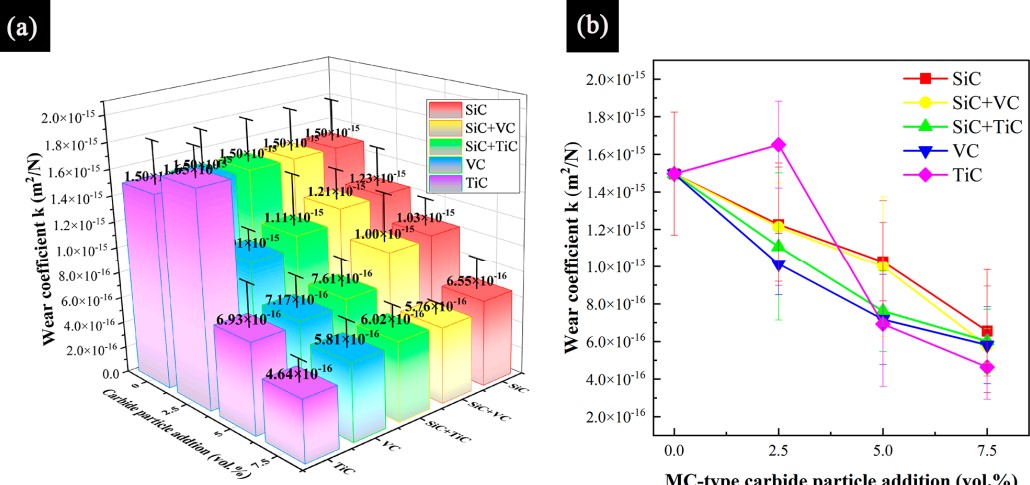

**Figure 11.** The effect of the volume fraction and type of additive MC-type carbides on the variation of κ wear coefficients of prepared HSS MMCs. (**a**) Histograms of wear coefficients vs. volume fraction and type of carbides; (**b**) Curves of wear coefficients vs. MC-type carbides addition.

The SEM and EDS micrographs of the worn pin surface of designed MMCs are exhibited in Figure 12. The grooves were observed on the worn surface, which indicates that abrasive wear is the main mechanism among the MMCs. The grooves were generated from the motion of abrasive particles detached from MMCs or counterparts on the pin surface. The worn pin surfaces exhibit grooving abrasive wear, which is mainly characterized by two-body abrasive wear or three-body abrasive wear with high loads [39]. An observation of additional carbides and EDS mapping results found that the existence of additional carbides could hinder grooving abrasive wear, ascribed to the higher hardness of carbides than the matrix. Meanwhile, the oxide debris and films generated from the high-speed wear test were observed behind the MC carbide. Notably, the type and size of additional carbides had a significant effect on the shape of the oxidized debris distribution on the worn pin surfaces. Compared to the MT-3 sample, the MS-3 and MV-3 samples presented a more homogeneous distribution of oxidized debris. Furthermore, the higher hardness and stability of TiC made it possible for the oxide to accumulate more easily behind the TiC. In addition, the distribution of carbides could also influence the morphology of oxidized debris [14,40,41]. Based on the above discussion, the wear mechanism of designed MMCs was a mixture of grooving abrasive wear and oxidation wear.

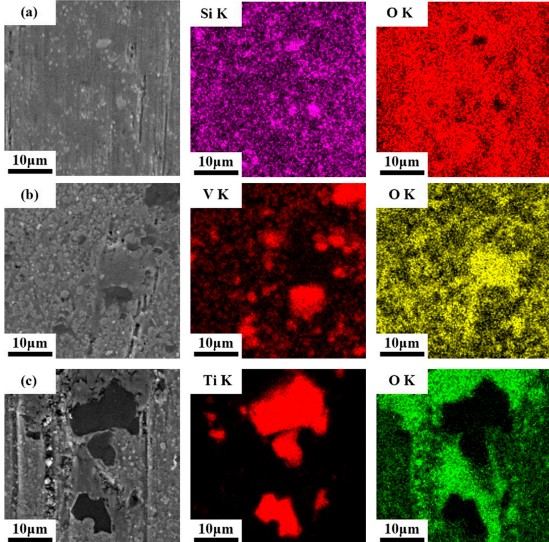

**Figure 12.** The SEM-SE micrographs of the worn MMCs pin surface and corresponding EDS mapping results. (**a**) MS-3, (**b**) MV-3, and (**c**) MT-3.

## 4. Conclusions

In this study, S390 MMCs were prepared using the SPS method, and the effect of MC-type carbides on wear coefficient, hardness, and toughness was also investigated. The following conclusions can be drawn:

- The density reduction in MMCs was proportionate to the volume fraction of MC-type carbides. The abnormal relative density enhancement could be attributed to the decomposition of SiC particles, which led to a reduction in the size of SiC carbides and the number density of pores. The layer of in situ carbides was observed around all MC-type carbides. The microstructure and formation mechanism of these in situ carbides were characterized and clarified. A (V, W)-rich phase precipitated around the TiC particle, acting as adhesion promoters between the matrix and the TiC particle.
- The hardness of MMCs was significantly influenced by the type of MC-type carbides. The MT-3 sample with TiC had the highest hardness (86.8 HRA) and exceeded the M sample by 1.8 HRA. The three-point bending strength was mainly influenced by the volume fraction of the MC-type carbide. The bending strength of the MMCs decreased from 1197 MPa to 899.3 MPa with TiC increasing from 0 to 7.5 vol.%.

- The designed MMCs in this study exhibited excellent wear resistance. Abrasive wear and oxidation wear were the primary mechanisms responsible for the wear failure of MMCs. The wear coefficients of MMCs were mainly influenced by the type and volume fraction of MC-type carbides. With an increase in the volume fraction, the wear coefficient decreased, which shows an evident improvement in wear resistance. The MT-3 MMC (specific wear coefficient of $4.64 \times 10^{-16}$ m$^2$/N) was approximately three times lower than the M sample (specific wear coefficient of $1.50 \times 10^{-15}$ m$^2$/N).

**Author Contributions:** Conceptualization, Q.H. and M.W.; methodology, Q.H. and H.L.; validation, Q.H. and H.L.; investigation, Q.H.; resources, Y.C.; data curation, Q.H.; writing—original draft preparation, Q.H.; writing—review and editing, H.L. and Z.S.; supervision, M.W.; project administration, M.W.; funding acquisition, M.W. All authors have read and agreed to the published version of the manuscript.

**Funding:** The authors acknowledge funding support from the National Natural Science Foundation of China, under grant number 51975240.

**Institutional Review Board Statement:** Not applicable.

**Informed Consent Statement:** Not applicable.

**Data Availability Statement:** All data are available from the corresponding author upon reasonable request.

**Conflicts of Interest:** The authors declare no conflict of interest.

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
