# Peer review of "The Effect of MC-Type Carbides on the Microstructure and Wear Behavior of S390 High-Speed Steel Produced via Spark Plasma Sintering"

_metals, doi:10.3390/met12122168_

Round 1

Reviewer 1 Report

Interesting research results and a good publication

Author Response

We also thank the reviewer for carefully review our paper. And reviewer’s comment encourage us to continue this study.

Reviewer 2 Report

The manuscript is studied on the microstructures and mechanical properties of the HSS reinforced with different volume fractions of MC-type carbides produced by SPS. It is interesting but it needed to be major revised. 

Author Response

Response to Reviewer #2

Thank the reviewer for these important suggestions to help us revise this manuscript. These questions were explained and revised one by one in the new manuscript.

Question 1: I wondered why the authors added MMCs up to 7.5wt.% because the hardness and wear properties were increased with the volume fraction of MMCs up to 7.5% and it could be increased much higher with the larger amount of MMCs.

Response: Thank you for the comment. Generally, particle reinforcement is widely used to improve the microstructures and properties of MMCs. And the type, size, and volume fraction of reinforcing particles are the important factors affecting the performance of MMCs. However, the comprehensive mechanical properties of MMCs are dependent on the matrix of MMCs and additional particles. Notably, S390 MMCs are required to have both excellent wear properties and toughness due to harsh in-service environments. As the reviewer thought, the more additional carbides could increase the hardness and wear properties, while also decrease the toughness of MMCs (such as a reduction of three-point bending strength in this work). Moreover, 7.5 vol.% is an upper value of additional carbide content according to prior studies. Therefore, it is not a good way for S390 MMCs to furtherly increase the volume fraction of additional carbides due to the deterioration of the toughness.

Question 2:During the heat treatment on the as-sintered samples in page 4, is there any reason two step heating was carried out for the small specimens? Also in the text, the authors mentioned that the triple tempering treatment was conducted at 560 C. But in the figure 2, it was wrote as 550 C.

Response: Thank you for the comment. Generally, the physical properties of alloys, such as thermal conductivity, are much dependent on their composition and microstructure. Especially for high-alloyed HSS steel, their thermal conductivity is much lower than that of a low-alloyed steel. The fast heating rates could lead to higher thermal stress which causes the cracking of as-sintered samples with an amount of porosity and extrinsic carbides in this study. Therefore, two-step heating is used here to reduce the cracking tendency.

For triple tempering treatments, this heating treatment was conducted at 550℃. The error has been revised in new papers.

Question 3: In the page 7, the authors mentioned that some black or dark grey carbides with a diameter of 5 are uniformly distributed around the powder boundary in carbide reinforced MMDs samples (Fig.4(b-c). But it could not be confirmed in the pictures because the figures were too bad to detect the carbides, pores and some defects.

Response: Thank you for the comment. Fig.4 has been replaced by the clear figure in the revised paper. And carbides and porosities have been labeled in new Fig.4.

Question 4: In the figure 5, the authors explained the presence of FeSi phase in SiC reinforced MMCs samples during the SPS process or heat treatment. But I am not sure the easy decomposition of SiC and the easy formation of FeSi. Also according to the XRD data, FeSi Peaks were also observed in Sample MV-1 where Si was not added.

Response: Thank you for the comment again. We have checked the XRD data of the MV-1 sample. The V2C phase (PDF #19-1393) has been labeled in new figure, and the existence of V2C might come from the powder of VC. New XRD data has been processed by Jade 6.0 and added to the revised paper. (Page 7. Fig.5)

Question 5: In the page 9, the authors explained that the addition of Ti C and VC could strikingly improve the hardness of the S930 MMCs but I am wondering that 1-2 values increasing from 86HRA could be the striking increased. It could be in the error dimension.

Response: Thank you for the comment. The sentence “As shown in Fig. 6, the addition of TiC and VC could strikingly improve the hardness of the S390 MMCs’’ was not appropriate in the prior paper. And this error was revised in a new paper: “As shown in Fig. 6, the addition of TiC and VC might help to improve the hardness of the S390 MMCs” (page 9. Line 262).

Reviewer 3 Report

The work describes the sintering of high-speed steel using SPS. MC carbides were added to improve mechanical properties and tribological performance.    Comments and suggestions are made as follows:

1. In the abstract, it was written sparkle. Please replace for spark.

2. Why were selected SiC, TiC, and VC? 

3. VC is expected to be present. How was VC from steel distinguished from VC from powder? 

4. What were the criteria for defining the volume fraction of MC? 

5. In figure 1, please add the information regarding whether it is secondary or backscattered electrons. 

6. What was the ball's material? Was it a mixture or material ground? 

7. SPS processing parameters, such as heating rate and vacuum, should be given.

8. Regarding figure 2, the holding time at 550 and 850 must be given. Is the tempering temperature 560 or 550C?

9. Figure 4 needs to be more explicit about the phases. It cannot be observed grain size. It can be observed a slight difference in the grayscale. Is it backscattered electrons? An EDS analysis should improve the evaluation. 

10. In Figure 5, there are many XDR peaks not identified. 

11. Pag 8, line 162, what do you mean by carbides precipitated at room temperature? 

12. Figures 6 and 7 are related; I suggest they are put together to make more accessible the analysis and comparison between equilibrium and non-equilibrium conditions 

13. Figure 7 needs to be in focus. 

14. Ti, V, and Mo form complex carbides. It will depend on when they were formed, such as solidification and heat treatment. Cabides addition and the carbides present in the alloy will influence the formation of the precipitates.  This work shows a fundamental approach regarding microstructural analysis and phase transformations but could be discussed more.   

15. The effect on bending and mechanical properties requires a statistical analysis. There are many variables evaluated in this work, and the approach was superficial

16. The wear test uses the SPed and heat-treated samples and a counter body of steel. 1000N is a high load. The analysis conducted is poor. It is not possible to see much.  Mechanical analysis and wear tests require o more detailed analysis.  

  In conclusion, the work presents essential results, which were not well evaluated since many results were given, but the discussion could have been better. Select the best results, and a statistical analysis may help show the contribution of the research done.       

Author Response

Response to Reviewer 3#

Thank the reviewer for these important suggestions to help us revise this manuscript. These questions were explained and revised one by one in the new manuscript.

Question 1: In the abstract, it was written sparkle. Please replace for spark.

Response: Thank you for the comment. The sparkle has been replaced by spark. (Title, Page 1. Line 11, and Page 1. Line 33).

Question 2: Why were selected SiC, TiC, and VC?

Response: Thank you for the comment. As shown in Table 6, the common physical characterization of these three carbides is the high hardness, which is beneficial in increasing the wear resistance and hardness of the MMCs. The chemical composition of VC is similar to that of intrinsic MC carbides in HSS matrix. And thus it is easy to bond with the matrix, leading to a less reduction in toughness. TiC is commonly used for producing cemented carbide cutting tool due to higher hardness than VC. Compared to SiC and VC, TiC is more helpful for improving the wear properties. Though hardness of SiC is lowest among three carbides, it could react with iron matrix to bind with each other, which is beneficial to reduce porosity. Therefore, these three carbides was selected in this study.(Page 3. Line 81)

Question 3: VC is expected to be present. How was VC from steel distinguished from VC from powder?

Response: Thank you for the comment. We can distinguish the VCs by their size and morphology. The VC from powder (5.83 μm from Table.3.) has larger diameter than VC from steel (1~2 μm from Fig.6(a)). The morphology of VC from powder (Fig.1(c), Fig.4(c) and Fig.7(b)) is more irregular than VC from steel (Fig.7(b)).

Question 4: What were the criteria for defining the volume fraction of MC?

Response: Thank you for the comment. The volume fraction of MC from powder was mainly determined according to other literatures (Processing of AISI M2 High-Speed Steel Reinforced with Vanadium Carbide by Solar Sintering, Wear Mechanisms in High-Speed Steel Reinforced with (NbC)p and (TaC)p MMCs). And the volume fraction of additional MC were basically in the range of 0-10 vol.%. Due to high volume fraction of intrinsic carbides in S390 matrix, volume fraction of extrinsic carbides were limited to below 7.5 vol.% to hold good toughness.  (Page 4. Line 102)

Question 5: In figure 1, please add the information regarding whether it is secondary or backscattered electrons.

Response: Thank you for the comment. The Fig.1 was captured in secondary electron mode. The information of SEM micrographs was added to new paper (Fig.1, Fig.4, Fig.6, Fig.10 and Fig.12).

Question 6: What was the ball's material? Was it a mixture or material ground?

Response: Thank you for the comment. The ball’s materials has been added to the revised paper. In revised paper, “The mixture of S390 powders and carbide particles was further mixed for 2 hours at 400 rpm with a ball-to-powder weight ratio of 7:1, and 316L stainless steel balls with a 5 mm diameter were employed to attain a uniform mixture.” (Page 4. Line 93)

Question 7: SPS processing parameters, such as heating rate and vacuum, should be given.

Response: Thank you for the comment. The heating rate was 90 ℃·min-1 and the SPS experiment was performed in non-vacuum state. The heating rate and vacuum state of SPS processing was added to the revised paper. (Page 4. Line 102)

Question 8: Regarding figure 2, the holding time at 550 and 850 must be given. Is the tempering temperature 560 or 550C?

Response: Thank you for the comment. The holding time at 550℃ and 850℃ is ~5 min, and the parameters have been labeled in new Fig.2. For triple tempering treatments, this heating treatment was conducted at 550℃. The error has been revised in new paper. (Page 4. Line 107 and Fig.2)

Question 9: Figure 4 needs to be more explicit about the phases. It cannot be observed grain size. It can be observed a slight difference in the grayscale. Is it backscattered electrons? An EDS analysis should improve the evaluation.

Response: Thank you for the comment. Figure 4 was replaced by clear micrographs where grain size and porosities could be observed. The micrographs in Figure 4 were captured under secondary electron mode, and the information was added to new paper. (Page 6. Line 170)

Question 10: In Figure 5, there are many XDR peaks not identified

Response: Thank you for the comment. The prior unidentified XRD peaks has been checked. After revise, the new figure was added to new paper. (Page 7. Fig.5 and Line 195)

Question 11: Pag 8, line 162, what do you mean by carbides precipitated at room temperature?

Response: Thank you for the comment. The expression has been rectified in revised paper. In revised paper, “As shown in Fig. 6(c), the MC carbide existed at high temperature has ~ 28 wt.% W elements, much higher than ones existed at room temperature.” (Page 8. Line 202)

Question 12: Figures 6 and 7 are related; I suggest they are put together to make more accessible the analysis and comparison between equilibrium and non-equilibrium conditions

Response: Thank you for the comment. According to your good advice, Figure 6 and Figure 7 are put together to become a new Figure 6. The new Figure 6 was added to new paper. (Page 7. Fig.6)

Question 13: Figure 7 needs to be in focus.

Response: Thank you for the comment. The micrographs in focus was added to new paper. (Page 7. Fig.6)

Question 14: Ti, V, and Mo form complex carbides. It will depend on when they were formed, such as solidification and heat treatment. Carbides addition and the carbides present in the alloy will influence the formation of the precipitates. This work shows a fundamental approach regarding microstructural analysis and phase transformations but could be discussed more.

Response: Thank you for the comment. The chemical composition of complex carbides could be influence by carbides addition and heat treatment procedure, especially by cooling rate, and the shape of carbide changed during isothermal holding for (Ti, Mo)C. The simulation results prove that (V,M)C (M = W, Mo and Cr) are thermodynamically stable and (Cr, Mo, W, Mn, V)-doped interface enhanced the fer-rite/TiC interface. The discussions and citations are added to revised paper to be more persuasive. (Page 9. Line 303)

  1. Zhang, K.; Wang, H.; Sun, X.J.; Sui, F.L.; Li, Z.D.; Pu, E.X.; Zhu, Z.H.; Huang, Z.Y.; Pan, H.B.; Yong, Q.-L. Precipitation Behavior and Microstructural Evolution of Ferritic Ti–V–Mo Complex Microalloyed Steel. Acta Metallurgica Sinica (English Letters). 2018, 31, 997–1005, doi:10.1007/s40195-018-0726-4.
  2. Wang, Z.; Zhang, H.; Guo, C.; Leng, Z.; Yang, Z.; Sun, X.; Yao, C.; Zhang, Z.; Jiang, F. Evolution of (Ti, Mo)C Particles in Austen-ite of a Ti–Mo-Bearing Steel. Mater. Des. 2016, 109, 361–366, doi:10.1016/j.matdes.2016.07.081.
  3. Jang, J.H.; Lee, C.H.; Han, H.N.; Bhadeshia, H.K.D.H.; Suh, D.W. Modelling Coarsening Behaviour of TiC Precipitates in High Strength, Low Alloy Steels. Mater. Sci. Tech. 2013, 29, 1074–1079, doi:10.1179/1743284713Y.0000000254.
  4. Sun, C.; Zheng, Y.; Chen, L.; Fang, F.; Zhou, X.; Jiang, J. Thermodynamic Stability and Mechanical Properties of (V, M)C (M = W, Mo and Cr) Multicomponent Carbides: A Combined Theoretical and Experimental Study. J. Alloys. Compd. 2022, 895, 162649, doi:10.1016/j.jallcom.2021.162649.
  5. Jang, J.H.; Lee, C.H.; Heo, Y.U.; Suh, D.W. Stability of (Ti,M)C (M=Nb, V, Mo and W) Carbide in Steels Using First-Principles Calculations. Acta. Mater. 2012, 60, 208–217, doi:10.1016/j.actamat.2011.09.051.
  6. Xiong, H.; Zhang, H.; Zhang, H.; Zhou, Y. Effects of Alloying Elements X (X=Zr, V, Cr, Mn, Mo, W, Nb, Y) on Ferrite/TiC Het-erogeneous Nucleation Interface: First-Principles Study. J. Iron. Steel. Res. Int. 2017, 24, 328–334, doi:10.1016/S1006-706X(17)30047-X.

Question 15: The effect on bending and mechanical properties requires a statistical analysis. There are many variables evaluated in this work, and the approach was superficial

Response: Thank you for the comment. The effect of content of carbides on mechanical properties was re-analyzed. And new analysis results have been added to the revised paper to show statistical analysis with standard deviation.

Question 16: The wear test uses the SPSed and heat-treated samples and a counter body of steel. 1000N is a high load. The analysis conducted is poor. It is not possible to see much.  Mechanical analysis and wear tests require o more detailed analysis.

Response: Thank you for the comment. The reason that a high load of 1000 N was applied in wear tests is that weight loss value of wear tests with a load of below 1000 N is too small to get reliable experimental data due to high hardness and good wear-resistance of MMCs. The effect of volume fraction on mechanical properties has been re-analyzed. And new results have been added to the revised paper.

Round 2

Reviewer 3 Report

The manuscript was improved, and corrections were made. I have no more comments. 

Author Response

Response to Reviewer 3#

We thank the reviewer for this valuable suggestion, and the whole manuscript has been polished accordingly.
